# Hardware-Based Activation Function-Core for Neural Network Implementations

Griselda González-Díaz_Conti [1,†], Javier Vázquez-Castillo [2,†], Omar Longoria-Gandara [3,†], Alejandro Castillo-Atoche [4,†], Roberto Carrasco-Alvarez [5,†], Adolfo Espinoza-Ruiz [1,†] and Erica Ruiz-Ibarra [1,*,†]

1 Electronics and Electrical Engineering Department, Technological Institute of Sonora, Ciudad Obregon 85000, Mexico; griselda.gonzalez18520@potros.itson.edu.mx (G.G.-D.); adolfo.espinoza@itson.edu.mx (A.E.-R.)
2 Department of Electrical Engineering, University of Quintana Roo, Chetumal 77019, Mexico; jvazquez@uqroo.edu.mx
3 Department of Electronics, Systems and IT, Western Institute of Technology and Higher Education, Tlaquepaque 45604, Mexico; olongoria@iteso.mx
4 Department of Mechatronics, Autonomous University of Yucatán, Merida 97000, Mexico; acastill@correo.uady.mx
5 Department of Electronics, University of Guadalajara, Guadalajara 44100, Mexico; r.carrasco@academicos.udg.mx
* Correspondence: erica.ruiz@itson.edu.mx; Tel.: +52-644-141-2650
† These authors contributed equally to this work.

**Abstract:** Today, embedded systems (ES) tend towards miniaturization and the carrying out of complex tasks in applications such as the Internet of Things, medical systems, telecommunications, among others. Currently, ES structures based on artificial intelligence using hardware neural networks (HNNs) are becoming more common. In the design of HNN, the activation function (AF) requires special attention due to its impact on the HNN performance. Therefore, implementing activation functions (AFs) with good performance, low power consumption, and reduced hardware resources is critical for HNNs. In light of this, this paper presents a hardware-based activation function-core (AFC) to implement an HNN. In addition, this work shows a design framework for the AFC that applies a piecewise polynomial approximation (PPA) technique. The designed AFC has a reconfigurable architecture with a wordlength-efficient decoder, i.e., reduced hardware resources are used to satisfy the desired accuracy. Experimental results show a better performance of the proposed AFC in terms of hardware resources and power consumption when it is compared with state of the art implementations. Finally, two case studies were implemented to corroborate the AFC performance in widely used ANN applications.

**Keywords:** artificial neural network; HW design framework; activation function; piecewise polynomial approximation; wordlength-efficient decoder

## 1. Introduction

Artificial neural networks (ANNs) are an important area of artificial intelligence (AI) used to perform several tasks, such as classification [1–4], pattern recognition [5–8], communications [9,10], control systems [11,12], prediction [13,14], among others. An ANN models a biological neural network employing a collection of nodes called artificial neurons, connected by edges to transmit signals like the synapses in a brain; during its transmission, the signal value changes according to the weight of the edges, adjusted by a learning process. Each artificial neuron processes the input signals through their weighted sum and the output through an activation function (AF), which can be non-linear. The neurons in an ANN are arranged into layers, and the signal travels from the first layer (input layer) to the last layer (output layer); between these layers, the signal can

travel through multiple internal layers (hidden layers). However, recent applications of ANNs, e.g., IoT, medical systems, and telecommunication, require platforms with high throughput and the capacity to execute the algorithms in real-time. An attractive solution is the development of hardware neuronal networks (HNN) in Field-Programmable Gate Arrays (FPGAs) [15–21]. In this regard, the FPGA-based implementation of AFs in HNN is one of the challenges for embedded system design according to recent studies; this is because the AF implementations require low hardware resources and low power consumption [1,2,5,12,22–25]. Currently, the most common non-linear functions for ANNs are the `Sigmoid` [11,26–32] ans `Tanh`AFs [22,32,33].

For instance, ref. [4] shows a convolutional neural network (CNN) that uses the `Tanh` AF in each layer, and ref. [22] presents a neuroevolution of augmenting topology, which employs the `Tanh` and `Gaussian` AFs in the hidden layer and output layer, respectively. On the other hand, the exponential linear unit (`ELU`) and softplus AFs are used for pattern classification CNNs as shown in [23,34], respectively.

In summary, the main contributions of this paper are:

1. A `Sigmoid`, hyperbolic tangent (`Tanh`), `Gaussian`, sigmoid linear unit (`SILU`), `ELU`, and `Softplus` AFs in reconfigurable hardware is designed with a piecewise polynomial approximation technique and a novel segmentation strategy.
2. A wordlength-efficient hardware decoder for an activation function-core (AFC) with a reduction in power consumption in the order of 13x gains in comparison with state-of-the-art works.
3. A design framework with the integration of an AFC to develop HNN applications.

The rest of the paper is organized as follows: the design methodologies for approaching AFs via PPA are presented in Section 2. The architecture and parameters for the AF hardware implementation are shown in Section 3. The hardware performance for the AFC using the proposed architecture is discussed in Section 4. The proposed AFC performance employing two case studies are presented in Section 5. Finally, conclusions are drawn in Section 6.

## 2. PPA Implementation Methodologies

Piecewise polynomial approximation (PPA) is a computing technique for the function approximation that offers a good trade-off between latency and memory resources. PPA splits the abscissa range into $K$ segments, considering the $x_i$ samples on an interval $[X_L, X_H]$ and the $f(x_i)$ function. In PPA, each one of the $K$ segments is approached by polynomial approximation as follows:

$$p_k(x_i) = a_n x_i^n + \cdots + a_1 x_i + a_0, \tag{1}$$

where $p_k(\cdot)$ are the polynomials corresponding to each segment, $k = 1, \cdots, K$; $a_n$ represents the polynomial coefficients, and $n$ stands the polynomial degree. In this sense, with the aim to evaluate PPA performance, the maximum absolute error (MAE), the mean squared error (MSE), and the mean absolute error (AAE) are proposed, which are given by

$$MAE = max_{(X_L \leq x_i \leq X_H)} |f(x_i) - p(x_i)|, \tag{2}$$

$$MSE = \frac{1}{N} \sum_{i=1}^{N} (f(x_i) - p(x_i))^2, \tag{3}$$

$$AAE = \frac{1}{N} \sum_{i=1}^{N} |f(x_i) - p(x_i)|, \tag{4}$$

where $p(\cdot)$ is the approximated function via PPA technique, and $N$ is the number of $x_i$ samples. However, the signal to quantization noise ratio (SQNR) is a metric to evaluate the performance in hardware applications based on fixed-point (FxP) arithmetic. SQNR is the ratio of the signal power of interest and the quantization noise power defined as follows

$$SQNR_{dB} = 10 \log_{10} \frac{\frac{1}{N} \sum_{i=1}^{N} f(x_i)^2}{\frac{1}{N} \sum_{i=1}^{N} (f(x_i) - p(x_i))^2}, \tag{5}$$

This study employs the PPA technique with a wordlength-efficient decoder (PPA-ED) methodology described in [35] to design the proposed AFC for HNN implementations. A comparative analysis is also provided to show the advantages of the proposed methodology with the minimax approximation [29], the simple canonical piecewise linear (SCPWL) [32], and the piecewise linear approximation computation (PLAC) [15].

### 2.1. Minimax Approximation

Minimax approximation minimizes the MAE across an input interval given an $n$-degree polynomial. This methodology takes into account the effect of rounding the coefficients to a finite wordlength, allowing a significant reduction in the size of the tables required to store the polynomial coefficients. Larkin et al. in [29] use the minimax technique to carry out the AF approximation, where a genetic algorithm is employed to obtain the segmentation, and a first-order approximation is applied:

$$a_1 x_i + a_0, \tag{6}$$

where $a_1$ and $a_0$ represent the slope and the constant term for a segment, respectively; they can be computed as presented in [36]. The proposal in [29] implements a reconfigurable hardware architecture for AF approximation. However, the polynomial indexation requires the whole input wordlength, which results in an excessive use of memory resources.

### 2.2. Simple Canonical Piecewise Linear

According to [32], SCPWL methodology has the ability to represent the non-linear AF behavior with low complexity and high speed. In this study, the AF approximation is described by

$$c_0 + \sum_{k=1}^{K} c_k \lambda_k(x_i), \tag{7}$$

where $k$ represents the segment index, $c_0$ is the constant term for all segments, and $c_k$ are the segment coefficients. However, the main disadvantage of SCPWL is that it requires a parallel execution of multiplications and sums to compute the contribution per segment for the polynomial evaluation of (7); consequently, the hardware resources are incremented.

### 2.3. Piecewise Linear Approximation Computation

PLAC is a methodology with an error-flattened segment that uses a linear approximation to improve the approximation performance under the desired MAE. This proposal splits the interval $[X_L, X_H]$ into discrete points given by

$$x_i \in \{X_L, X_L + \frac{1}{2^{iw}}, X_L + \frac{2}{2^{iw}}, \cdots, X_H\}, \tag{8}$$

where $iw$ represents the number of fractional bits for $x_i$. The number of discrete points, i.e., the number of $x_i$ samples is given by $N = (X_H - X_L)/2^{-iw} + 1$, where $i = 1, \cdots, N$. The coefficients for the first-order polynomial approximation, e.g., by using (6) in the segment $[x_a, x_b]$, can be computed by

$$a_1 = \frac{f(x_b) - f(x_a)}{x_b - x_a}, \tag{9}$$

$$a_0 = f(x_a) - a_1 x_a, \tag{10}$$

where $[x_a, x_b] \in x_i$ and $x_a < x_b$.

The open literature presents AF hardware implementations based on PLAC; e.g., ref. [15] proposes an architecture that reduces the polynomial indexation. An advantage of PLAC is that it applies a strategy for reducing the polynomial indexation. However, disadvantages of PLAC include its use of linear approximations for approaching the AF segments that increase the segments needed to achieve the desired MAE and the need for a large amount of memory resources.

### 2.4. PPA with Wordlength-Efficient Decoder

The PPA-ED [35] optimizes the polynomial indexation and improves previous methodologies [15,29,32] for the hardware design of AFs according to the MAE, MSE, AAE, and SQNR metrics. This study customizes the PPA-ED to design AFCs in HNN implementations on FPGAs. Figure 1 illustrates the methodology for designing an AFC with reduced hardware and improved performance for HNN design based on the proposed method.

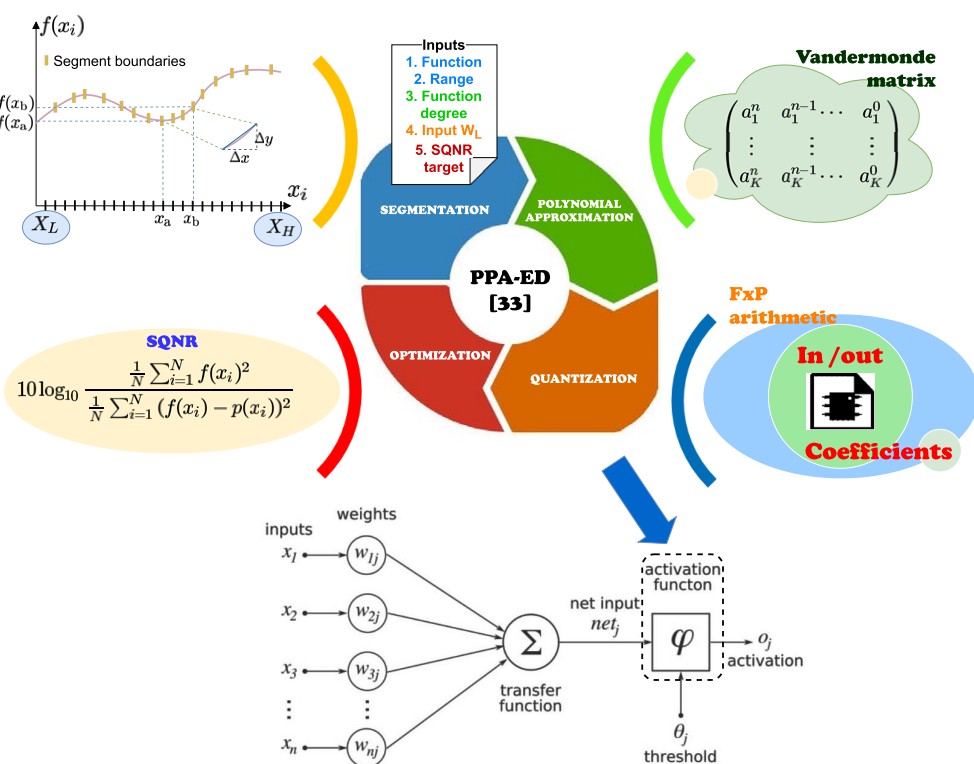

**Figure 1.** Methodology for designing the AFC.

The adaptive segmentation, the polynomial approximation, the quantization, and the optimization processes of PPA-ED reduce hardware resources used in the design. The used methodology automatically sets up the PPA segment limits, computed as linear combinations of power-of-two, using a detection algorithm based on the function slope and the user-defined parameters in order to achieve the desired performance and optimize the polynomial indexation [35]. Likewise, PPA-ED computes the $n+1$ coefficients of $n$-degree for the $K$ segments applying the Vandermonde Matrix [37]. The proposed methodology also considers a quantization process for hardware implementation that defines the FxP format required for guaranteeing the desired accuracy in terms of SQNR. All these features allow for the evaluation of the function with efficient polynomial indexing for the desired precision; consequently, a hardware architecture for the AFC with a wordlength-efficient decoder that reduces hardware resource usage is achieved.

### 3. AFC Hardware Implementation

The AFC implementation employs a design framework based on the proposed methodology [35]. Figure 2 shows the design framework for implementing the AFC in HNN. This study implements the AFs `Sigmoid`, `Tanh`, `Gaussian`, `SILU`, `ELU`, and `Softplus`, whose mathematical expressions are shown in Table 1. The framework consists of sequential processing stages for the AFC custom design on FxP arithmetic and its integration in HNN implementations. The workflow describes the design steps for a HW design on FPGA devices.

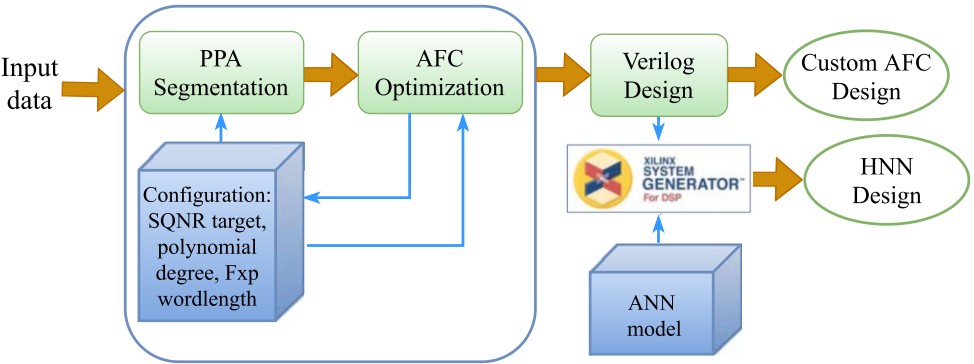

**Figure 2.** Design framework for implementing AFC in HNN.

**Table 1.** Description of non-linear AFs for ANN.

| AF | Mathematical Description | Symmetry | Evaluation Range |
|---|---|---|---|
| 1. `Sigmoid` | $f(x) = \frac{1}{1+e^{-x}}$ | $f(x) = \begin{cases} f(x) & x \geq 0 \\ 1 - f(\|x\|) & x < 0 \end{cases}$ | $(-8, 8)$ |
| 2. `Tanh` | $f(x) = \frac{e^x - e^{-x}}{e^x + e^{-x}}$ | $f(x) = \begin{cases} f(x) & x \geq 0 \\ -f(\|x\|) & x < 0 \end{cases}$ | $(-8, 8)$ |
| 3. `Gaussian` | $f(x) = e^{-x^2}$ | $f(x) = \begin{cases} f(x) & x \geq 0 \\ f(\|x\|) & x < 0 \end{cases}$ | $(-8, 8)$ |
| 4. `SILU` | $f(x) = \frac{x}{1+e^{-x}}$ | - | $(-8, 8)$ |
| 5. `ELU` | $f(x) = \begin{cases} \alpha(e^x - 1) & x \leq 0 \\ x & x > 0 \end{cases}$ | - | $(-4, 4)$ |
| 6. `Softplus` | $f(x) = \ln(1 + e^x)$ | - | $(-4, 4)$ |

The first stage of the framework is related to the PPA-ED configuration, e.g., polynomial degree, SQNR target, and FxP wordlength requirements. The second stage is the PPA segmentation, which calculates the optimum number of segments and the required coefficients for the function evaluation with the desired accuracy.

In the third stage, the PPA-ED optimizes the AFC in an iterate way. The fourth stage generates the custom Verilog design of the AFC. The last stage corresponds to the HNN design, which is obtained by using the designed AFC into a ANN model.

The AFC implementation uses the proposed reconfigurable and wordlength-efficient hardware architecture shown in Figure 3, where $a_n$ represents the polynomial coefficients, x is the input represented on FxP, and f(x) is the output of the evaluated function. Figure 4 depicts the function evaluator block for the proposed architecture, which computes a

second-order AF evaluation on FxP arithmetic by employing Horner's rule [36]. The LUT block contains the coefficients for AF evaluation, and the address decoder unit indexes the LUT according to the bits_agu frame shown in Figure 5. The LUT address has a width of $L = \lceil \log_2(K) \rceil$ bits, where K represents the number of segments and $\lceil \cdot \rceil$ stands for the ceil function. The proposed AFC architecture considers and exploits the AF symmetry to reduce the hardware resources.

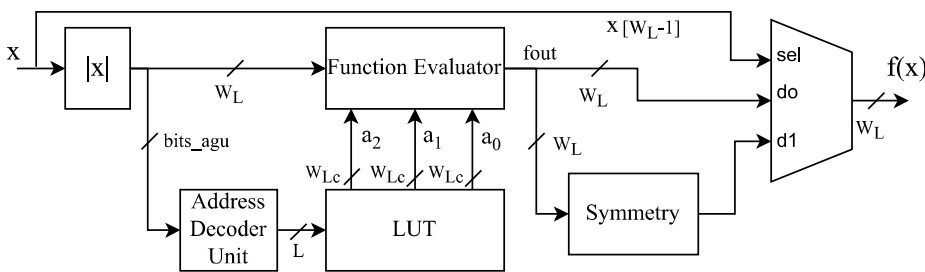

**Figure 3.** Proposed hardware architecture for implementing the AFC.

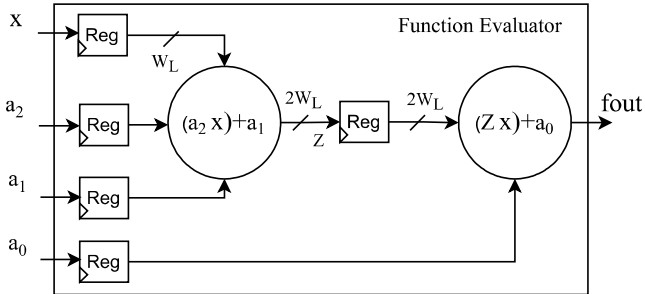

**Figure 4.** Function evaluator for the implemented AFC.

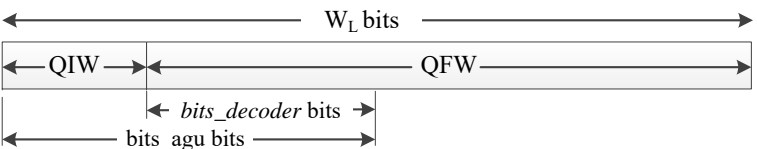

**Figure 5.** Input data structure, x.

Table 2 shows the FxP format corresponding to the input/output values and the polynomial coefficients. The FxP signed format $Q(W_L, QFW, s)$ considers a wordlength $W_L$ and the fractional bits QFW. In the case of the polynomial coefficients, the FxP signed format $Q(W_{L_C}, QFC, s)$ considers a $W_{L_C}$ wordlength and the fractional bits QFC.

**Table 2.** FxP format for implementing AFC in HNN.

| | Activation Function | Input/Output FxP Format $Q(W_L, QFW, s)$ | Coefficients FxP Format $Q(W_{L_C}, QFC, s)$ |
|---|---|---|---|
| 1. | Sigmoid | (16,10,s) | (16,15,s) |
| 2. | Tanh | (16,10,s) | (16,14,s) |
| 3. | Gaussian | (16,10,s) | (16,14,s) |
| 4. | SILU | (16,11,s) | (16,13,s) |
| 5. | ELU | (16,12,s) | (16,15,s) |
| 6. | Softplus | (16,12,s) | (16,15,s) |

## 4. Experimental Results and Discussion

To verify the proposed reconfigurable architecture for the AFC (see Figure 3), the selected AFs (see Tables 1 and 2) were approximated employing the proposed methodology in Section 2.4. Likewise, the implementation for the AFC architecture was according to the hardware specification and the polynomial coefficients shown in Tables 2–8. As was mentioned in the hardware design framework, Verilog is the hardware description language for implementing the AFC, synthesized on a Xilinx Artix-7 xc7a100t-2csg324 FPGA device. Tables 9–14 show the performance comparisons for the PPA-ED-based AFC implemented. In this sense, Table 9 shows the AFC performance results based on the methodologies minimax approximation [29] and PPA-ED, for the `Sigmoid` and `Tanh` AFs. The input/output Fxp representations were configured according to [29] for a fair comparison. As can be seen, for the `Sigmoid` AF with 4 segments, the proposed methodology improves [29], reducing the MAE in 55.3% and the AAE in 65.4%. Even increasing the segments to 6 in [29], PPA-ED reduces the MAE in 8.7% and the AAE in 30.8%. Likewise, for the case of `Tanh` AF with 4 segments, MAE is reduced in 37.9% and the AAE in 50%.

**Table 3.** Floating-point and fixed-point coefficients for implementing `Sigmoid` AF in HNN.

| Segment Number | Segment Boundaries | Format | $a_2$ | $a_1$ | $a_0$ |
|:---:|:---:|:---:|:---:|:---:|:---:|
| 1 | [0 1.5) | ★ | −0.0364 | 0.2685 | 0.4981 |
|   |         | ☐ | 0xfb55 | 0x225f | 0x3fc2 |
| 2 | [1.5 3.5) | ★ | −0.03 | 0.2243 | 0.551 |
|   |           | ☐ | 0xfc29 | 0x1cb4 | 0x4687 |
| 3 | [3.5 4.5) | ★ | −0.0086 | 0.0873 | 0.7712 |
|   |           | ☐ | 0xfee4 | 0x0b2b | 0x62b8 |
| 4 | [4.5 8.0) | ★ | −0.0012 | 0.0174 | 0.9356 |
|   |           | ☐ | 0xffd9 | 0x023b | 0x77c0 |

★ Floating-point. ☐ Fixed-point in hexadecimal notation.

**Table 4.** Floating-point and fixed-point coefficients for implementing `Tanh` AF in HNN.

| Segment Number | Segment Boundaries | Format | $a_2$ | $a_1$ | $a_0$ |
|:---:|:---:|:---:|:---:|:---:|:---:|
| 1 | [0 1) | ★ | −0.3269 | 1.0968 | −0.0055 |
|   |       | ☐ | 0xeb14 | 0x4631 | 0xffa5 |
| 2 | [1 2) | ★ | −0.1691 | 0.7027 | 0.2318 |
|   |       | ☐ | 0xf52d | 0x2cf8 | 0x0ed5 |
| 3 | [2 3.5) | ★ | −0.0189 | 0.1242 | 0.7933 |
|   |         | ☐ | 0xfeca | 0x07f2 | 0x32c4 |
| 4 | [3.5 8) | ★ | −0.0001 | 0.0019 | 0.9939 |
|   |         | ☐ | 0xfffd | 0x001f | 0x3f9b |

★ Floating-point. ☐ Fixed-point in hexadecimal notation.

**Table 5.** Floating-point and fixed-point coefficients for implementing `Gaussian` AF in HNN.

| Segment Number | Segment Boundaries | Format | $a_2$ | $a_1$ | $a_0$ |
|---|---|---|---|---|---|
| 1 | [0 0.50) | ⋆ | −0.9458 | −0.0065 | 1.0001 |
|   |          | □ | 0xc378 | 0xff96 | 0x4001 |
| 2 | [0.50 1.00) | ⋆ | −0.5053 | −0.2634 | 1.0377 |
|   |             | □ | 0xdfa9 | 0xef23 | 0x426a |
| 3 | [1.00 1.25) | ⋆ | 0.1554 | −1.0796 | 1.2914 |
|   |             | □ | 0x09f2 | 0xbae8 | 0x52a5 |
| 4 | [1.25 1.75) | ⋆ | 0.4353 | −1.6126 | 1.5452 |
|   |             | □ | 0x1bdc | 0x98ca | 0x62e5 |
| 5 | [1.75 2.50) | ⋆ | 0.3366 | −1.3284 | 1.3409 |
|   |             | □ | 0x158a | 0xaafc | 0x55d1 |
| 6 | [2.50 3.75) | ⋆ | 0.1318 | −0.6063 | 0.7035 |
|   |             | □ | 0x0870 | 0xd932 | 0x2d05 |
| 7 | [3.75 5.25) | ⋆ | 0.0157 | −0.0895 | 0.1281 |
|   |             | □ | 0x0100 | 0xfa45 | 0x0833 |
| 8 | [5.25 8.00) | ⋆ | 0 | −0.0001 | 0.0002 |
|   |             | □ | 0x0000 | 0xffff | 0x0002 |

⋆ Floating-point. □ Fixed-point in hexadecimal notation.

**Table 6.** Floating-point and fixed-point coefficients for implementing `SILU` AF in HNN.

| Segment Number | Segment Boundaries | Format | $a_2$ | $a_1$ | $a_0$ |
|---|---|---|---|---|---|
| 1 | [−8 −4.5) | ⋆ | −0.0045 | −0.0686 | −0.2641 |
|   |           | □ | 0xffda | 0xfdcd | 0xf78c |
| 2 | [−4.5 −2) | ⋆ | −0.0149 | −0.174 | −0.53002 |
|   |           | □ | 0xff86 | 0xfa6e | 0xef09 |
| 3 | [−2 −1) | ⋆ | 0.0798 | 0.2053 | −0.1453 |
|   |         | □ | 0x028d | 0x0691 | 0xfb5a |
| 4 | [−1 0.5) | ⋆ | 0.2329 | 0.4997 | 0.0015 |
|   |          | □ | 0x0773 | 0x0ffd | 0x000c |
| 5 | [0.5 2) | ⋆ | 0.115 | 0.6885 | −0.0689 |
|   |         | □ | 0x03ae | 0x1608 | 0xfdcb |
| 6 | [2 3.5) | ⋆ | −0.0095 | 1.1447 | −0.4912 |
|   |         | □ | 0xffb1 | 0x24a1 | 0xf047 |
| 7 | [3.5 6) | ⋆ | −0.0115 | 1.1431 | −0.4611 |
|   |         | □ | 0xffa1 | 0x2494 | 0xf13e |
| 8 | [6 8] | ⋆ | −0.0024 | 1.039 | −0.1635 |
|   |       | □ | 0xffec | 0x213f | 0xfac4 |

⋆ Floating-point. □ Fixed-point in hexadecimal notation.

**Table 7.** Floating-point and fixed-point coefficients for implementing `ELU` AF in HNN.

| Segment Number | Segment Boundaries | Format | $a_2$ | $a_1$ | $a_0$ |
|---|---|---|---|---|---|
| 1 | $[-4 \; -2.5)$ | ★ | 0.004 | 0.0345 | −0.1227 |
|   |   | □ | 0x0084 | 0x046c | 0xf04a |
| 2 | $[-2.5 \; -1.5)$ | ★ | 0.0138 | 0.0831 | −0.0621 |
|   |   | □ | 0x01c4 | 0x0aa1 | 0xf80e |
| 3 | $[-1.5 \; -0.5)$ | ★ | 0.0375 | 0.1507 | −0.0133 |
|   |   | □ | 0x04cd | 0x134a | 0xfe4d |
| 4 | $[-0.5 \; 0)$ | ★ | 0.0783 | 0.1961 | −0.0001 |
|   |   | □ | 0x0a05 | 0x191a | 0xfffc |

★ Floating-point. □ Fixed-point in hexadecimal notation.

**Table 8.** Floating-point and fixed-point coefficients for implementing `Softplus` AF in HNN.

| Segment Number | Segment Boundaries | Format | $a_2$ | $a_1$ | $a_0$ |
|---|---|---|---|---|---|
| 1 | $[-4 \; -2)$ | ★ | 0.0238 | 0.1948 | 0.4184 |
|   |   | □ | 0x030c | 0x18ef | 0x358e |
| 2 | $[-2 \; 0)$ | ★ | 0.0969 | 0.472 | 0.68844 |
|   |   | □ | 0x0c67 | 0x3c68 | 0x581e |
| 3 | $[0 \; 2)$ | ★ | 0.0969 | 0.528 | 0.68844 |
|   |   | □ | 0x0c67 | 0x4397 | 0x581e |
| 4 | $[2 \; 4]$ | ★ | 0.0238 | 0.8052 | 0.4184 |
|   |   | □ | 0x030c | 0x6710 | 0x358e |

★ Floating-point. □ Fixed-point in hexadecimal notation.

In order to compare the performance between the PPA-ED-based AFC and SCPWL implementation [32], the proposed methodology was configured with a number of segments to provide a similar number of polynomial coefficients according to the implementation results reported in [32]. Likewise, the AFC architectures were configured with the same FxP requirements. Table 10 shows the AFC performance comparison implemented via PPA-ED and SCPWL. As the implementation results show, the proposed method reduces the MAE in 59% and 61%, and the MSE in 48% and 67%, for the `Sigmoid` and `Tanh` AFs, respectively. Likewise, comparison results for architectures designed via PLAC [15] and PPA-ED methodologies can be observed in Table 11. In this case, the PPA-ED-based AFC reduces the MAE in 30.79% and 29.54% for `Sigmoid` and `Tanh` AFs, respectively.

Table 12 shows the power consumption results of the PPA-ED-based AFC designs obtained by Xilinx Power Analyzer. These results are outstanding compared with the other proposals; e.g., Table 13 shows the power comparison results when implementing the `Sigmoid` AF via Minimax approximation [29]. As can be seen, the AFC implemented via PPA-ED improves the average power consumption at least 13x. Finally, Table 14 shows the hardware resources used for implementing PPA-ED-based AFCs according to the proposed design framework. The AFCs achieve a maximum work frequency of 51.71 MHz, 53.69 MHz, 59.45 MHz, 57.87 MHz, 57.64 MHz, and 57.74 MHz for Sigmoid, Tanh, Gaussian, ELU, SILU, and Softplus, respectively.

As can be seen in Tables 9–14, experimental results have shown a better performance of the PPA-ED methodology to implement AFCs in terms of MAE, MSE, AAE, SQNR, and power consumption, achieving a power reduction of at least 13x for the `Sigmoid` AF.

**Table 9.** Performance comparison for the implemented AFs based on minimax approximation and PPA-ED methodologies.

| | | $x \in$ Q(14,10,s), $f(x) \in$ Q(12,10,s) | | | | |
|---|---|---|---|---|---|---|
| **Function** | **Proposal** | **Segments** | **SQNR [dB]** | **Range** | **MAE** | **AAE** |
| Sigmoid | Larkin * [29] | 4 | NA | $(-8, 8)$ | $4.7 \times 10^{-3}$ | $2.4 \times 10^{-3}$ |
| | | 6 | NA | | $2.3 \times 10^{-3}$ | $1.2 \times 10^{-3}$ |
| | PPA-ED ** | 4 | 59.49 | | $2.1 \times 10^{-3}$ | $8.3 \times 10^{-4}$ |
| Tanh | Larkin * [29] | 4 | NA | $[0, 8)$ | $9.5 \times 10^{-3}$ | $2.4 \times 10^{-3}$ |
| | PPA-ED ** | 4 | 53.40 | | $5.9 \times 10^{-3}$ | $1.2 \times 10^{-3}$ |
| | | 3 | 50.60 | | $10.7 \times 10^{-3}$ | $1.7 \times 10^{-3}$ |

* First order polynomial. ** Second order polynomial.

**Table 10.** Performance comparison for the implemented AFs based on SCPWL and PPA-ED methodologies.

| | | x, f(x) $\in$ Q(16,10,s) | | | | |
|---|---|---|---|---|---|---|
| **Function** | **Proposal** | **Segments** | **SQNR [dB]** | **Range** | **MAE** | **MSE** |
| Sigmoid | Hussein * [32] | 9 | NA | $(-8, 8)$ | $5.2 \times 10^{-3}$ | $1.8 \times 10^{-6}$ |
| | PPA-ED ** | 4 | 56.76 | | $2.1 \times 10^{-3}$ | $9.2 \times 10^{-7}$ |
| Tanh | Hussein * [32] | 9 | NA | $(-8, 8)$ | $15.4 \times 10^{-3}$ | $1.2 \times 10^{-5}$ |
| | PPA-ED ** | 4 | 53.55 | | $5.9 \times 10^{-3}$ | $3.9 \times 10^{-6}$ |
| Gaussian | Hussein * [32] | 9 | NA | $(-8, 8)$ | $7.0 \times 10^{-3}$ | $1.4 \times 10^{-5}$ |
| | PPA-ED ** | 8 | 49.48 | | $1.7 \times 10^{-3}$ | $8.9 \times 10^{-7}$ |
| | PPA-ED | 6 | 41.96 | | $3.9 \times 10^{-3}$ | $5.23 \times 10^{-6}$ |

* First order polynomial. ** Second order polynomial.

**Table 11.** Performance comparison for the implemented AFs based on PLAC and PPA-ED methodologies.

| | | x, f(x) $\in$ Q(8,8,ns) | | | |
|---|---|---|---|---|---|
| **Function** | **Proposal *** | **Segments** | **Range** | **SQNR [dB]** | **MAE** |
| Sigmoid | Dong [15] | 2 | $[0, 1)$ | NA | $5.65 \times 10^{-3}$ |
| | PPA-ED | 2 | | 48.22 | $3.91 \times 10^{-3}$ |
| Tanh | Dong [15] | 4 | $[0, 1)$ | NA | $5.55 \times 10^{-3}$ |
| | PPA-ED | 4 | | 46.03 | $3.91 \times 10^{-3}$ |

* First order polynomial.

**Table 12.** Hardware performance for the implemented PPA-ED-based AFC.

| **Function** | **Range** | **SQNR** | **MAE** | **AAE** | **Power Consumption * [mW]** |
|---|---|---|---|---|---|
| Sigmoid | $(-8, 8)$ | 56.76 | $2.1 \times 10^{-3}$ | $9.2 \times 10^{-7}$ | 0.82 |
| Tanh | $(-8, 8)$ | 53.55 | $5.9 \times 10^{-3}$ | $3.9 \times 10^{-6}$ | 0.88 |
| Gaussian | $(-8, 8)$ | 53.55 | $5.9 \times 10^{-3}$ | $3.9 \times 10^{-6}$ | 0.88 |
| ELU | $(-4, 4)$ | 78.73 | $5.6 \times 10^{-4}$ | $1.1 \times 10^{-4}$ | 0.53 |
| SILU | $(-8, 8)$ | 60.14 | $7.9 \times 10^{-3}$ | $2.6 \times 10^{-3}$ | 0.96 |
| Softplus | $(-4, 4)$ | 59.50 | $5.2 \times 10^{-3}$ | $1.4 \times 10^{-3}$ | 0.66 |

* Frequency 40 MHz.

**Table 13.** Power consumption comparison for Sigmoid AF.

| Proposal | Segments | MAE | AAE | Frequency [MHz] | Average Power [mW] |
|---|---|---|---|---|---|
| Larkin [29] | 8 | $1.3 \times 10^{-3}$ | $0.9 \times 10^{-3}$ | 40 | 17 |
| PPA-ED-based AFC | 5 | $1.2 \times 10^{-3}$ | $3.6 \times 10^{-4}$ | 40<br>50 | 1.02<br>1.27 |

**Table 14.** Hardware resource usage for the PPA-ED-based AFCs.

| HW Resources | Consumption by Function | | | | | | | Available | Utilization % |
|---|---|---|---|---|---|---|---|---|---|
| | Sigmoid | Tanh | Gaussian | ELU | SILU | Softplus | | | |
| Slice register | 1 | 1 | 0 | 16 | 0 | 0 | Out of | 126,800 | 0% |
| Slice LUTs | 76 | 78 | 93 | 28 | 54 | 22 | Out of | 63,400 | 0% |
| IOBs | | | 33 | | | | Out of | 210 | 15% |
| BUFG/ BUFGCTRLs | | | 1 | | | | Out of | 32 | 3% |
| DSP48E1s | | | 2 | | | | Out of | 240 | 0% |

## 5. Hardware Neural Networks: Case Studies

Two case studies on ANN applications support the implemented PPA-ED-based AFCs, which were selected because ANNs are continuously under research and the development of devices considering reduced hardware has relevance for the applications of embedded systems based on HNNs [38–43]. The efficiency of the proposed PPA-ED methodology is demonstrated on an FPGA-based accelerator(AFC), which employs minimal hardware resources. Here, under the co-simulation paradigm, the validation of the proposed design was conducted. In this sense, an FxP-based reference ANN (golden model) was developed using Matlab/Simulink, and the results were compared with a design according to the proposed design framework for HNNs (see Figure 2).

The first case study is related to the implementation of a digital classification neural network that uses an ELU AFC. The digit classification neural network identifies digits between zero and nine. The second case study uses a Tanh AFC to implement a breast cancer neural network application, the aim of which is to classify the cancers as either benign or malignant depending on the characteristics of sample biopsies.

### 5.1. Digit Classification

The digital classification applied the MNIST database of handwritten digits with a training matrix of 60,000 rows × 785 columns [44]. Each row of the matrix represents one digit of the database containing a label and image of 28 × 28 pixel grayscale; the digit label is in the first column, and the remaining 784 columns have the pixel information.

Figure 6 shows the implemented digit classification ANN structure. In this case study, 50 epochs were computed for the training process. The simulation results show that the PPA-ED-based AFC has an error of 0.01%, identifying 97.2% of the samples, which converge to the results generated by the FxP-based reference ANN (golden model).

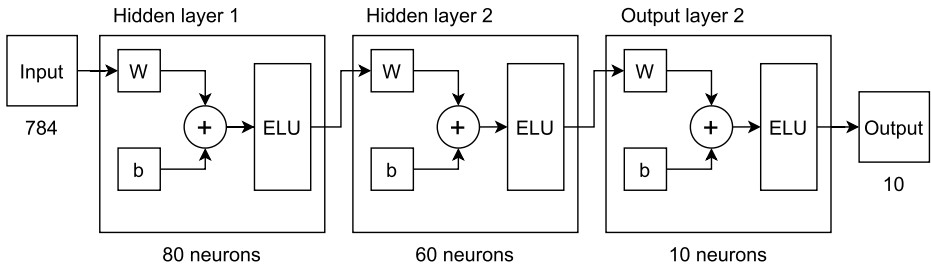

**Figure 6.** Digit classification ANN structure.

### 5.2. Breast Cancer Detection

The case study on breast cancer detection applied the data set provided by Matlab, with an input matrix with 9 rows × 699 columns [45]. The columns represent the biopsies with the attributes contained in the rows. The breast cancer detection ANN structure has ten neurons in one hidden layer and two in the output layer. The hidden layer employs the `Tanh` AFC. This ANN computed 27 epochs for the training process.

The performance comparison for the breast cancer ANN is shown in Figure 7, in which the ANN placed at the top corresponds to the ANN implemented by Simulink blocks, and the ANN at the bottom side corresponds to the proposed PPA-ED-based design under co-simulation. Both models achieved an accuracy of 97.80%, which demonstrates the effectiveness of the proposed model.

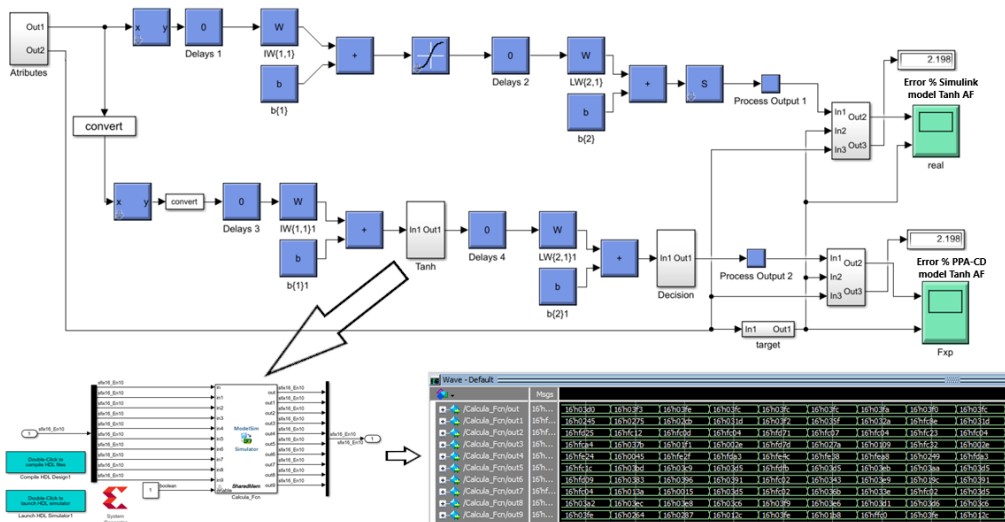

**Figure 7.** Breast cancer detection ANN performance comparison.

## 6. Conclusions

In this paper, the use of the PPA-ED methodology to implement AFC in HNN was presented. The proposal is focused on the AFC implementation providing an efficient architecture and configuration parameters; however, the tune on of the ANN hyperparameters is offline. In order to reach this aim, a reconfigurable and wordlength-efficient decoder for the AFC hardware architecture was proposed. This architecture performs a second-order polynomial function evaluation to approach the selected AFs. In this sense, a hardware neural network framework was introduced, which allows verifying the proposed PPA-ED-based design in terms of the MAE, AAE, and SQNR metrics. Likewise, a comparative analysis was provided to show the advantages of the PPA-ED in contrast to the minimax approximation, SCPWL, and PLAC methodologies. Additionally, two case studies were presented to corroborate the AFC in widely used ANN applications. Finally, experimental results have shown a better performance of the PPA-ED methodology to implement AFCs in terms of MAE, AAE, SQNR, and power consumption, achieving a power reduction of at least 13x for the `Sigmoid` AF. AFC performance analysis on floating-point arithmetic is considered for further works.

**Author Contributions:** Conceptualization, G.G.-D., J.V.-C., A.C.-A. and O.L.-G.; formal analysis, G.G.-D., J.V.-C., O.L.-G., A.C.-A. and R.C.-A.; investigation, G.G.-D. and J.V.-C.; methodology, G.G.-D., J.V.-C., O.L.-G. and A.E.-R.; project administration, J.V.-C. and E.R.-I.; validation, O.L.-G., A.C.-A., A.E.-R. and E.R.-I.; writing—original draft, G.G.-D., J.V.-C. and O.L.-G.; writing—review and editing, G.G.-D., J.V.-C., O.L.-G., A.C.-A., R.C.-A., A.E.-R. and E.R.-I. All authors have read and agreed to the published version of the manuscript.

**Funding:** This research was funded by the Instituto Tecnológico de Sonora through PROFAPI projects number 2021_0092 and 2021_0116. The APC was funded by PROFAPI projects number 2021_0092 and 2021_0116.

**Conflicts of Interest:** The authors declare no conflict of interest.

**Sample Availability:** The data will be made available under request.

## Abbreviations

The following abbreviations are used in this manuscript:

| | |
|---|---|
| AAE | mean absolute error |
| AI | Artificial intelligent |
| ANN | Artificial neural network |
| AF | Activation function |
| AFC | Activation function-core |
| CNN | Convolutional neural network |
| dB | Decibels |
| ELU | Exponential linear unit |
| FPGA | Field programmable gate arrays |
| FxP | Fixed point |
| MAE | Maximum absolute error |
| MSE | mean squared error |
| PLAC | Piecewise linear approximation computation |
| PPA | Piecewise polynomial approximation |
| PPA-ED | PPA with wordlength-efficient decoder |
| SILU | Sigmoid linear unit |
| SCPWL | Simple canonical piecewise linear |
| SQNR | Signal to quantization noise ratio |
| Tanh | Hyperbolic tangent |
| HNN | Hardware neural network |
| HW | Hardware |

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
