# Peer review of "Hardware-Based Activation Function-Core for Neural Network Implementations"

_electronics, doi:10.3390/electronics11010014_

Round 1

Reviewer 1 Report

The manuscript entitled “Hardware-based activation function-core for neural network implementations” aims to reduce power consumption and hardware resources required by neural networks by proposing: different non-linear activation functions based on piecewise polynomial approximation and segmentation; a reconfigurable architecture and a wordlength-efficient decoder for an activation function-core; and a design framework for implementation. The manuscript is well written and engaging, providing very interesting information for the readers. The significance of the study and the presentation quality are high. All the sections are consistent and rigorous; the results are clearly presented. However, there are a few minor issues that should and could easily be addressed prior to publication:

1.    In the Authors’ names, please check the orcid links.
2.    In the case studies section, please be more specific in the criteria you used to choose the applications, for example adding references to lines 192-193 “widely used ANN applications” or shortly explaining why these cases are useful for evaluations.
3.    Line 197 “Matlab” and 215 “MATLAB”, please check capitalization consistency.
4.    Please provide references for both the data sets that you used in the case studies section by citing the publication in which the data were originally published.
e.g., for the MNIST dataset: 
Y. LeCun, L. Bottou, Y. Bengio, and P. Haffner. "Gradient-based learning applied to document recognition." Proceedings of the IEEE, 86(11):2278-2324, November 1998. 
5.    The conclusions section could be improved by including one or two sentences about the study limitations and explicit recommendations for further research or applications.

I recommend the publication after this minor revision.

Author Response

  1.  In the Authors’ names, please check the orcid links.

All ORCID links were updated.

  1. In the case studies section, please be more specific in the criteria you used to choose the applications, for example adding references to lines 192-193 “widely used ANN applications” or shortly explaining why these cases are useful for evaluations.

Many thanks for your comment, the section “Hardware Neural Network: Cases of Studies” was modified (from 198 to 201 lines) in order to be more specific, as follows:

Two cases-of-studies of ANN applications support the implemented PPA-ED-based AFCs, which were selected because ANN is continuously under research and the development of devices considering reduced hardware takes relevance for embedded system's applications based on HNN [39-44].

Added references:

39. Ahlawat, S., Choudhary, A., Nayyar, A., Singh, S., Yoon, B. (2020). Improved Handwritten Digit Recognition Using Convolutional Neural Networks (CNN). Sensors 2020, 20, 3344. https://doi.org/10.3390/s20123344

40. Alwzwazy, H. A., Albehadili, H. M., Alwan, Y. S., Islam, N. E. (2016). Handwritten digit recognition using convolutional neural networks. International Journal of Innovative Research in Computer and Communication Engineering, 4(2), 1101-1106.

41. Ali, S., Shaukat, Z., Azeem, M., et al. (2019). An efficient and improved scheme for handwritten digit recognition based on convolutional neural network. SN Appl. Sci. 1, 1125. https://doi.org/10.1007/s42452-019-1161-5

42. Fung Fung Ting, Yen Jun Tan, Kok Swee Sim. (2019). Convolutional neural network improvement for breast cancer classification. Expert Systems with Applications, Volume 120, Pages 103-115. https://doi.org/ 10.1016/j.eswa.2018.11.008.

43. Alom, M.Z., Yakopcic, C., Nasrin, M.S., et al. (2019). Breast Cancer Classification from Histopathological Images with Inception Recurrent Residual Convolutional Neural Network. J Digit Imaging 32, 605–617. https://doi.org/10.1007/s10278-019-00182-7

44. Mesut Toǧaçar, Kutsal Baran Özkurt, Burhan Ergen, Zafer Cömert. (2020). BreastNet: A novel convolutional neural network model through histopathological images for the diagnosis of breast cancer. Physica A: Statistical Mechanics and its Applications, Volume 545. https://doi.org/10.1016/j.physa.201

  1. Line 197 “Matlab” and 215 “MATLAB”, please check capitalization consistency.

Both words were verified and corrected in the revised version of the manuscript.

  1. Please provide references for both the data sets that you used in the case studies section by citing the publication in which the data were originally published. e.g., for the MNIST dataset:

LeCun, L. Bottou, Y. Bengio, and P. Haffner."Gradient-based learning applied to document recognition." Proceedings of the IEEE, 86(11):2278-2324, November 1998.

Thanks for the suggestions, were added the references for the MNIST database of handwritten digits in the first paragraph of the Digit Classification subsection and the breast cancer database in the first paragraph of the Breast Cancer Detection subsection, as follows:

Line 213:

The digital classification applied the MNIST database of handwritten digits with a training matrix of 60,000 rows × 785 columns [45]….

Line 222:

 The case of study about breast cancer detection applied the data set provided by MATLAB, with an input matrix with 9 rows × 699 columns [46]….

 References:

 45. Available online: URL (accessed on 25 September 2021) Langelaar J. (2021). MNIST neural network training and testing. MATLAB Central File Exchange.

46. Murphy, P.M., Aha, D.W. UCI Repository of machine learning databases. [http://www.ics.uci.edu/ mlearn/MLRepository.html]. Available online: URL (accessed on 25 September 2021)

  1. The conclusions section could be improved by including one or two sentences about the study limitations and explicit recommendations for further research or applications.

Thank you for the suggestion. The conclusion section was improved with the following two sentences: the first sentence in line 234 considers the delimitation for the work, and the second one in line 247 shows a recommendation.

Line 234:

The proposal is focused on the AFC implementation providing an efficient architecture and configuration parameters; however, the tune on of the ANN hyperparameters is offline.

Line 247:

AFC performance analysis on floating-point arithmetic is considered for further works.

Reviewer 2 Report

The manuscript describes a hardware block to implement various activation functions to be used in artificial neural networks.

The paper is clear, however I have a few amendments.

More details of the architecture in Fig. 3 must be given, especially the inner content of the Function Evaluator. 

Is it possible to give the maximum achievable clock frequencies for the various described implementations?

Is it possible to add comparisons with other state-of-the-art works?

Author Response

1. More details of the architecture in Fig. 3 must be given, especially the inner content of the Function Evaluator. 

A new Figure 4 was added according to the reviewer’s suggestion (see lines from 143 to 147).  Figure 3 was also updated to identify the output of the function evaluator block.

The AFC implementation uses the proposed reconfigurable and wordlength-efficient hardware architecture shown in Figure 3, where an represents the polynomial coefficients, x is the input represented on FxP, and f(x) is the output of the evaluated function. Figure 4 depicts the function evaluator block for the proposed architecture, which computes a second-order AF evaluation on FxP arithmetic by employing Horner’s rule [34]….

2. Is it possible to give the maximum achievable clock frequencies for the various described implementations?

Thanks for your suggestion. The maximum clock frequency was described (lines 190 to 192) as follows:

The AFCs achieve the maximum work frequency of 51.71 MHz, 53.69 MHz, 59.45 MHz, 57.87 MHz, 57.64 MHz, 57.74 MHz for Sigmoid, Tanh, Gaussian, ELU, SILU, Softplus, respectively.

3. Is it possible to add comparisons with other state-of-the-art works?

Thanks for the reviewer's suggestion. Two new references (see lines 32 and 35) were incorporated in the revised version of the manuscript.

References:

19. Bouguezzi, et al. (2021). An Efficient FPGA-Based Convolutional Neural Network for Classification: Ad-MobileNet. Electronics 2021, vol. 10, no. 18, 2272. https://doi.org/10.3390/electronics10182272

23. Kim, et. al. (2021). AERO: A 1.28 MOP/s/LUT Reconfigurable Inference Processor for Recurrent Neural Networks in a Resource-Limited FPGA.Electronics 2021, vol. 10, no. 11, 1249. https://doi.org/10.3390/electronics10111249

Reviewer 3 Report

In this study, the authors present a hardware-based activation function-core (AFC) to implement a hardware neural network. Although the idea looks ok, there still have some major concerns related to the use cases, model evaluation,... Some detailed comments are as follows:

1. The authors evaluated the efficiency of their method on 2 simple datasets, thus the contribution is still limited. At least it should be validated on a more complicated dataset to see their performance.

2. More discussions should be added to discuss the results/findings.

3. When comparing the performance among methods, did all of them use the same dataset and evaluation method?

4. Statistical tests should be conducted when comparison to show the significant differences among methods/models.

5. Uncertainties of model should be reported.

6. Deep learning or ANN has been used in previous studies i.e., PMID: 34812044, PMID: 31380767. Thus, the authors are suggested to refer to more works in this description to attract a broader readership.

7. How did the authors tune the optimal hyperparameters of the model?

8. English language should be improved.

Author Response

1. The authors evaluated the efficiency of their method on 2 simple datasets, thus the contribution is still limited. At least it should be validated on a more complicated dataset to see their performance.

Thanks for your reviewing work and for the opportunity of providing an answer to your concerns. However, the validation does not depend on a specific database. Rather, the efficiency mainly relies on the speed and hardware resource metrics. To clarify this concern, the contributions of the paper were re-edited highlighting the design of AF hardware neural networks.

The main contributions (see lines 43 to 49) of the paper are the following:

  • A Sigmoid, hyperbolic tangent Tanh, Gaussian, sigmoid linear unit (SILU), ELU, and Softplus AFs in reconfigurable hardware is designed with a piecewise polynomial approximation technique and a novel segmentation strategy.
  • A wordlength-efficient hardware decoder for an activation function-core (AFC) with a reduction of power consumption in the order of 13x gains in comparison with the state-of-the-art works.
  • A design framework with the integration of an AFC to develop HNN applications.

The performance of the proposed PPA-ED-based design is verified in terms of the MAE, MSE, AAE, and SQNR metrics and comparing the results (MAE, MSE, AAE, and SQNR) with state-of-the-art architectures.

According to the HNN Design framework in Figure 2, the proposed architectures are efficient according to the flexibility to configure parameters and the achieved reduced hardware design in comparison to other state-of-art architectures.

The ANN dataset was used only to corroborate the ANN performance.

2. More discussions should be added to discuss the results/findings.

Many thanks for your suggestion. The authors have incorporated a brief discussion highlighting our main results in Section 4 “Experimental Results and Discussion” (see lines 193-196). Also, a description of the comparative analysis from Table 9 to Table 14 was incorporated.

3. When comparing the performance among methods, did all of them use the same dataset and evaluation method?

Thanks for the comment. Traditionally hardware architectures are tested in terms of speed and area resources. As described above, the evaluation with the MAE, MSE, AAE, and SQNR metrics was highlighted through the revised version of the manuscript.

4. Statistical tests should be conducted when comparison to show the significant differences among methods/models.

The results of the proposal (designed architecture) are verified in terms of the MAE, MSE, AAE, and SQNR, as traditionally used on HW design.  Lines 203 to 206 describe the verification process to corroborate that the results of the proposed AFC converge with the golden model results (or ANN model developed in SW).

5. Uncertainties of model should be reported.

Thanks again for the opportunity to clarify our contribution. Our contribution is not a model or a function library. Our challenge is developing an optimized HW architecture for ANN applications on reconfigurable embedded devices. The design for the AFC should comply with the configuration restrictions and metrics taking like reference the design framework. We did not find uncertainties in the proposed design framework.

6. Deep learning or ANN has been used in previous studies i.e., PMID: 34812044, PMID: 31380767. Thus, the authors are suggested to refer to more works in this description to attract a broader readership.

To motivate the reader, the following eight relevant references were added throughout the paper. Two references correspond to the following cites added in the first paragraph of Introduction (line 20)

  1. Tng, SS, Le, NQK. Yeh, HY. Chua, MCH. (2021). Improved Prediction Model of Protein Lysine Crotonylation Sites Using Bidirectional Recurrent

Neural Networks. J Proteome Res. (2021). doi: 10.1021/acs.jproteome.1c00848.

  1. Le, NQ, Nguyen, BP. (2019). Prediction of FMN Binding Sites in Electron Transport Chains based on 2-D CNN and PSSM Profiles. IEEE/ACM Transactions on Computational Biology and Bioinformatics. DOI: 10.1109/tcbb.2019.

And the remaining references were introduced in the first paragraph (lines 198-201) of the Hardware Neural Network: Cases of Studies section, as follows:

Two cases-of-studies of ANN applications support the implemented PPA-ED-based AFCs, which were selected because ANN is continuously under research and the development of devices considering reduced hardware takes relevance for embedded system's applications based on HNN [39-44].

Added references:

  1. Ahlawat, S., Choudhary, A., Nayyar, A., Singh, S., Yoon, B. (2020). Improved Handwritten Digit Recognition Using Convolutional Neural Networks (CNN). Sensors 2020, 20, 3344. https://doi.org/10.3390/s20123344

  1. Alwzwazy, H. A., Albehadili, H. M., Alwan, Y. S., Islam, N. E. (2016). Handwritten digit recognition using convolutional neural networks. International Journal of Innovative Research in Computer and Communication Engineering, 4(2), 1101-1106.

  1. Ali, S., Shaukat, Z., Azeem, M., et al. (2019). An efficient and improved scheme for handwritten digit recognition based on convolutional neural network. SN Appl. Sci. 1, 1125. https://doi.org/10.1007/s42452-019-1161-5

  1. Fung Fung Ting, Yen Jun Tan, Kok Swee Sim. (2019). Convolutional neural network improvement for breast cancer classification. Expert Systems with Applications, Volume 120, Pages 103-115. https://doi.org/ 10.1016/j.eswa.2018.11.008.

  1. Alom, M.Z., Yakopcic, C., Nasrin, M.S., et al. (2019). Breast Cancer Classification from Histopathological Images with Inception Recurrent Residual Convolutional Neural Network. J Digit Imaging 32, 605–617. https://doi.org/10.1007/s10278-019-00182-7

  1. Mesut Toǧaçar, Kutsal Baran Özkurt, Burhan Ergen, Zafer Cömert. (2020). BreastNet: A novel convolutional neural network model through histopathological images for the diagnosis of breast cancer. Physica A: Statistical Mechanics and its Applications, Volume 545. https://doi.org/10.1016/j.physa.201

7. How did the authors tune the optimal hyperparameters of the model?

Even when the research about topologies and tune methodologies for the hyperparameter in the ANN is interesting, this paper proposes an activation function-core for implementing any ANN, easing the hardware implementation in an ANN.

8. English language should be improved.

Thanks for your suggestion. The revised version of the manuscript was completely revised and corrected.

Round 2

Reviewer 3 Report

My previous comments have been addressed well.